DATA RELEASE

# A chromosome-level reference genome for Pacific herring (*Clupea pallasii)* from the Bering Sea

Laura E. Timm[1,2,*], Yin-Chen Hsieh[1], J. Andrés López[1,3], Sydney A. Almgren[1] and Jessica R. Glass[1]

1 College of Fisheries and Ocean Sciences, University of Alaska Fairbanks, 2150 Koyukuk Drive, 99775, Fairbanks, USA

2 Resource Ecology and Fisheries Management Division, NOAA NMFSC Alaska Fisheries Science Center, Seattle, 98115, USA

3 Museum of the North, University of Alaska Fairbanks, 1962 Yukon Drive, 99775, Fairbanks, USA

## ABSTRACT

Pacific herring (*Clupea pallasii*) serve as a critical trophic link between plankton and many marine species targeted by fisheries. With a broad distribution throughout the North Pacific Ocean, from the Arctic to temperate latitudes, herring hold ecological, economic, and cultural importance. Despite this importance, genomic resources for this species, such as reference genome sequences, have only recently become available. To date, only one scaffold-level reference genome, representing a specimen from the Gulf of Alaska (Vancouver; 1,379 scaffolds), has been published to NCBI. Addressing this data gap, we produced a high quality 795 Mb genome sequence organized into 26 chromosomes combining long read sequencing with short read sequencing of proximity ligation libraries. Our assembly is highly complete (BUSCO score of 97.7%) and contiguous (922 contigs, N50 = 7,338,470, L50 = 38; 26 scaffolds, N50 = 31,494,017; L50 = 12). Pacific herring from the Bering Sea are genetically differentiated from those south of the Aleutian Islands and the Alaska Peninsula, making a reference genome from the eastern Bering Sea an important addition to the Pacific herring's genomic toolbox.

**Subjects** Genetics and Genomics, Evolutionary Biology, Molecular Genetics

**Submitted:** 27 March 2026

\* Corresponding author. Email: letimm@alaska.edu

Preprint submitted at https://doi.org/10.64898/2026.02.09.704930

## CONTEXT

Forage fishes are small pelagic species that consume plankton and serve as prey items for seabirds, marine mammals, and several fishes [1, 2]. Forage fishes, with their characteristic boom-and-bust abundance cycles, are especially vulnerable to the impacts of climate change [3]: ten years ago, a powerful marine heatwave resulted in a massive and extended disruption of forage fish abundance in the North Pacific [4]. As marine heatwaves increase in duration and severity, climate resilient forage fish fisheries are critical for maintaining ecosystem services and food security, regionally and globally [5, 6]. Management of forage fish fisheries, including stock identification, assessments, and evaluation of their temperature-based adaptability or survivability, depends on the availability of accurate and comprehensive genetic data for forage fish species. However, data gaps persist for forage fishes, including a paucity of reference genomes [7, 8].

From the subtropics to the Arctic, Pacific herring (*Clupea pallasii*) is one of the most abundant forage fish species in the Pacific Ocean. Recent whole genome resequencing research targeting *C. pallasii* in the Northeast Pacific identified a biogeographic break along

**Table 1.** Feature comparisons between publicly available chromosome level assemblies from species of Clupeiformes and the Clupal_KotzSound assembly presented here.

| Genomic features | *Clupea pallasii* (Kotzebue) | *Clupea pallasii* (British Columbia) | *Clupea harengus* | *Sprattus sprattus* | *Alosa alosa* | *Alosa sapidissima* | *Sardina pilchardus* |
|---|---|---|---|---|---|---|---|
| Assembly ID | Clupal_Kotz Sound | GCA_05117654 5.1_fCluPal1_p1.1 | GCF_90070041 5.2_Ch_v2.0.2 | GCA_96345772 5.1_fSprSpr1.1 | GCF_01758949 5.1_AALO_Geno_1.1 | GCF_01849268 5.1_fAloSap1.pri | GCF_96385418 5.1_fSarPil1.1 |
| Total length of assembled contigs (Mb) | 815.0 | 851.8 | 785.4 | 840.3 | 854.4 | 903.6 | 869.4 |
| Number of contigs | 922 | 1,379 | 3,713 | 1,887 | 2,630 | 1,712 | 1,894 |
| Number of scaffolds | 26 | 1,379 | 1,723 | 386 | 1,094 | 73 | 240 |
| Number of chromosomes | 26 | 26 | 26 | 21 | 24 | 24 | 24 |
| Contig N50 (Mb) | 7.3 | 1.7 | 1.0 | 1.2 | 1.2 | 1.6 | 1.1 |
| Scaffold N50 (Mb) | 31.5 | 1.7 | 29.8 | 33.8 | 35.4 | 38.4 | 34.6 |
| Contig L50 | 38 | 142 | 198 | 186 | 207 | 165 | 218 |
| Scaffold L50 | 12 | 142 | 13 | 9 | 12 | 11 | 12 |
| GC content | 44% | 44% | 44% | 44% | 43% | 43% | 45% |
| Coverage | 69.4× | 46.2× | 75× | 62× | 140× | 33× | 33× |

the Alaska Peninsula and Aleutian Islands, as well as cryptic diversity in the eastern Bering Sea [9]. However, due to the lack of a chromosome-level reference genome for Pacific herring, this work has relied on the reference genome of its sister species, Atlantic herring (*Clupea harengus*), to map short reads. This practice risks confounding population genomic analyses and demographic inferences, misidentifying outlier loci, and misclassifying the size and position of structural variants [10–13]. To address this data gap, we produced a chromosome-level assembly of *C. pallasii* from Kotzebue, Alaska, USA, referred to as the Clupal_KotzSound assembly.

The Clupal_KotzSound reference genome represents a crucial resource for the further study of *C. pallasii* and related species within Clupeiformes in higher latitude regions. This reference genome is based on *de novo* long reads aligned to scaffolds generated with proximity ligation and has a size of approximately 795 Mb arranged in 26 autosomes. Chromosomes range in size from 14.84 Mb to 40.25 Mb. The total estimated genome size and chromosome number of Clupal_KotzSound is consistent with whole genome assemblies of other forage fish species, specifically *C. harengus, Sprattus sprattus*, *Alosa alosa*, *Alosa sapidissima*, and *Sardina pilchardus*, while constructed of fewer contigs assembled into chromosome-scale scaffolds (Table 1). To compare chromosome evolution between clupeid assemblies, we aligned Clupal_KotzSound to the chromosome-level reference *C. harengus* assembly and an existing, scaffold-level *C. pallasii* genome collected from British Columbia. These pairwise genome alignments revealed an overall conservation of chromosomal arrangements, with some notable inversions and restructuring between *C. harengus* and *C. pallasii*, and potentially within the two *C. pallasii* genomes themselves.

Here, we present the first chromosome-level reference genome for *C. pallasii* and a comparative analysis of reference genomes publicly available for Clupeiformes. While comparisons of genomic architecture within *C. pallasii* are precluded by the number of scaffolds constituting the British Columbia assembly, we do find evidence of substantial genetic distance between the two assemblies, as well as chromosomes dominated by inversions between the *C. harengus* and Clupal_KotzSound assemblies. The Clupal_KotzSound assembly is a high-quality genomic resource to support critical future research on herring and other clupeiform fishes.

## APPROACH

### Sample acquisition

Two spawning female herring were collected by a subsistence fisher in Kotzebue, Alaska, USA in June 2024 and donated for sequencing. Whole specimens were immediately stored in a dry shipper containing liquid nitrogen and shipped to the Museum of the North at the University of Alaska Fairbanks, where they were transferred to storage at –80 °C. First, liver tissue samples were collected from each individual and shipped to Phase Genomics (Seattle, Washington, USA) and the whole specimens were moved to a –20 °C freezer. Heart tissue samples were also taken from each individual and were later shipped to Phase Genomics to increase the yield of high molecular weight DNA and intact cells.

### Data generation

Genomic DNA was prepared for chromatin conformation capture with the Proximo Hi-C (Animal) 4.0 Kit, following the manufacturer's protocol [14]. Generally, DNA within intact cells was crosslinked with a formaldehyde solution prior to digestion with restriction enzymes (DPNII, Ddel, Hinfl, and Msel). Proximity ligation with biotinylated nucleotides resulted in chimeric molecules, composed of genomic regions that were physically proximal *in vivo*. Biotinylated molecules were captured on streptavidin beads and prepared for Illumina sequencing with the Next Ultra DNA kit (New England Biolabs). Sequencing on an Illumina NovaSeq generated 260,583,196 PE150 read pairs.

High molecular weight DNA was extracted for PacBio sequencing using the PacBio NanoBind Pan DNA kit (Circulomics) and fragmented according to the Megaruptor 3 DNA Shearing Guide (Diagenode). Following fragmentation, the sequencing library was prepared with the PacBio SMRTbell prep kit 3.0 using the SMRTbell barcoded adapter plate 3.0, and the Revio Polymerase kit was used for binding and cleanup. Sequencing occurred on the PacBio Revio, as well as demultiplexing, quality control, and adapter trimming with Lima v2.12.0. PacBio sequencing generated 55,021,844,244 bps across 10,537,187 reads.

### Sequence assembly and scaffolding

Phased haplotype assembly drafts were built from Hi-C and PacBio Hifi reads in hifiasm v0.19.9 with default parameters [15]. Following the Phase Genomics Proximo Hi-C kit recommendations, Hi-C reads were aligned to the hifiasm assembly using the BWA-MEM algorithm v0.7.17 [16], skipping pairing (–P) and mate rescue (–S), marking the smallest coordinate as primary in the case of split alignment (–5), and parallelizing across eight threads (–t 8). PCR duplicates were flagged in SAMBLASTER v0.1.24 [17] and excluded from downstream analysis. All secondary and non-primary alignments were filtered (–F 2304) in samtools v1.22 [18].

The single-phase scaffolding procedure was followed to generate chromosome-scale scaffolds on the Phase Genomics' Proximo Hi-C genome scaffolding platform: a contact frequency matrix was calculated from aligned Hi-C read pairs and normalized by the restriction site count associated with each contig [19]. Scaffolds were constructed to optimize expected contact frequency [20] and scaffolding errors were manually corrected in Juicebox v2.3.6 [21, 22].

**Table 2.** Clupal_KotzSound assembly statistics.

| Type | Contig (bp) | Scaffold (bp) |
|---|---|---|
| Total number | 922 | 26 |
| L50 | 38 | 12 |
| N10 | 15,141,956 | 34,812,277 |
| N50 | 7,338,470 | 31,494,017 |
| N90 | 680,145 | 27,395,609 |
| Max length | 22,522,838 | 40,256,301 |
| Total length | 815,006,063 | 792,619,606 |

## Genome assembly validation

QUAST v. 5.2.0 [23] was used to summarize the quality metrics of the Clupal_KotzSound genome assembly compared to two other herring reference genomes currently on NCBI: *C. harengus* (Ch_v2.0.2) [24] and *C. pallasii* from British Columbia - "BC" (fCluPal1_p1.1) [25]. QUAST comparisons were computed first without a reference genome, then twice using *C. harengus* and *C. pallasii* BC as references. The Clupal_KotzSound genome assembly consists of 922 contigs organized into 26 scaffolds and 296 unaligned contigs, including a contig representing the complete mitochondrial genome (ptg000670c__unscaffolded) and a chimeric mitochondrial contig (ptg000700l__unscaffolded was removed during NCBI Genome processing). The assembly spans a total genome length of 815.1 Mb, a size falling in between that of the current reference *C. harengus* assembly at 786.3 Mb and reference *C. pallasii* BC genome assembly of 851.8 Mb. When compared to both of these reference assemblies, Clupal_KotzSound is constructed over fewer scaffolds (323 compared to 1496 in *C. harengus* and 1380 in *C. pallasii* BC), achieving a contig N50 of 7.3 Mb and a scaffold N50 of 31.5 Mb (Table 2). The scaffold N50 of Clupal_KotzSound is comparable to the 29.8 Mb scaffold N50 of the *C. harengus* assembly and is considerably higher than N50 of 1.7 Mb in the *C. pallasii* BC assembly. The GC content of Clupal_KotzSound is 44%, consistent with GC content in *C. harengus* and *C. pallasii* BC. From these QUAST contiguity metrics, our Clupal_KotzSound assembly meets or exceeds the quality range of existing herring reference genome assemblies.

## COMPARISON BETWEEN EXISTING CLUPEIFORM GENOMES

## Genome content analysis

The Clupal_KotzSound assembly completeness was analyzed by Benchmarking Universal Single-Copy Orthologs (BUSCO) v6.0.0 with the actinopterygii_odb12 lineage dataset [26, 27]. BUSCO analysis compared Clupal_KotzSound with six published genomes: *Clupea pallasii* BC [25], *C. harengus* [24], *Sprattus sprattus* [28], *Alosa alosa* [29], *A. sapidissima* [30], and *Sardina pilchardus* [31] (Tables 1, 3). Single-copy complete coding sequences were extracted from assemblies with Another Gff Analysis Toolkit (AGAT) v0.7.0 [32] and aligned with Multiple Alignment of Coding SEquences (MACSE) v2 [33, 34]. Across available genomes, missingness ranges from 1.48% (*A. sapidissima*) to 5.52% (*A. alosa*). Clupal_KotzSound is the second most complete of those analyzed (98.15%), falling 0.03% below *A. sapidissima* and 2.15% above the scaffold-level assembly of *C. pallasii* BC. 95.84% of the 7,207 ortholog clusters derived from 75 Actinopterygii were identified in Clupal_KotzSound, 12.73% higher than the British Columbia assembly.

Repeat content was assessed for the Clupal_KotzSound, *C. pallasii* BC, and *C. harengus* assemblies in RepeatMasker v4.1.9 [35] with dfam release 3.9. Clupal_KotzSound had the

**Table 3.** Percent loci from the actinopterygii_odb12 found in publicly available chromosome level assemblies from species of Clupeiformes and the scaffold-level assembly from a specimen of *Clupea pallasii* sampled in British Columbia. The actinopterygii_odb12 gene set consists of 7207 ortholog clusters derived from 75 ray finned fish genome assemblies.

| Species | Assembly ID | Complete (%) | Single-copy (%) | Duplicated (%) | Fragmented (%) | Missing (%) |
|---|---|---|---|---|---|---|
| *Clupea pallasii* KS | This assembly | 98.15 | 95.84 | 2.32 | 0.31 | 1.54 |
| *Clupea harengus* | GCF_900700415.2_Ch_v2.0.2 | 96.27 | 92.95 | 3.32 | 1.30 | 2.43 |
| *Clupea pallasii* BC | GCA_051176545.1_fCluPal1_p1.1 | 96.00 | 83.11 | 12.89 | 0.90 | 3.09 |
| *Sprattus sprattus* | GCA_963457725.1_fSprSpr1.1 | 98.04 | 96.70 | 1.35 | 0.29 | 1.67 |
| *Alosa alosa* | GCF_017589495.1_AALO_Geno_1.1 | 93.94 | 92.51 | 1.43 | 0.54 | 5.52 |
| *Alosa sapidissima* | GCF_018492685.1_fAloSap1.pri | 98.18 | 97.02 | 1.17 | 0.33 | 1.48 |
| *Sardina pilchardus* | GCF_963854185.1_fSarPil1.1 | 98.09 | 96.42 | 1.67 | 0.25 | 1.67 |

highest proportion of repeat content and the *C. pallasii* BC assembly had the lowest (13.9% and 12.7%, respectively). However, this may reflect the fragmented nature of the British Columbia assembly, as repetitive regions are difficult to reliably sequence.

## Pairwise genome alignment and synteny

Pairwise whole genome alignments of Clupal_KotzSound and *C. harengus* and *Sprattus sprattus* were constructed with minimap2 [36] in D-GENIES [37] and the resulting pairwise mapping format files were visualized as dot plots (Supplemental Figure 1). Pairwise alignment of Clupal_KotzSound with *C. harengus* revealed genome-wide, large-scale structural similarities, with good agreement between the two species on the overall arrangement, size, and number of chromosomes, with some large inversions and other structural shifts (Supplemental Figure 1A). Comparing Clupal_KotzSound and *Sprattus sprattus* required aligning different numbers of chromosomes (26 and 21, respectively) and while some chromosomes aligned relatively cleanly, large structural differences were identified across the alignment (Supplemental Figure 1B). Pairwise alignment of Clupal_KotzSound and *C. pallasii* BC showed much less conclusive agreement between the genomes and substantial gaps in the alignment, largely due to the scaffold-level resolution of the *C. pallasii* BC assembly, which precluded visualization. These whole genome alignments point to shared global genomic arrangements between *C. pallasii* and *C. harengus*, potential intra-specific rearrangements between *C. pallasii* BC and *C. pallasii* from Kotzebue Sound (KS), and a need for higher resolution analyses to identify intra-chromosomal genetic structures.

To investigate fine-scale differences in genomic structure, a pairwise genome alignment limited to chromosome-length scaffolds was generated with minimap2 [36] and genomic synteny between the *C. harengus* assembly and Clupal_KotzSound was analyzed with the Synteny and Rearrangement Identifier (SyRI) v1.7.1 [38] assuming low divergence between genomes. At the sequence level, 2.76 million single nucleotide polymorphisms were identified, as well as 0.53 million indels (Table 4). We identified 1,899 syntenic regions and 407 inversions (Figure 1). While no chromosome was completely free of inversions, several chromosomes were dominated by large inversions: three were identified on chromosome 6, ranging from ~1.61 Mb to ~2.68 Mb, the longest of which has nearly identical breakpoints to those defining a homologous inversion in Atlantic herring [39, 40]; one on chromosome 7, ~6.03 Mb; one near the beginning of chromosome 16, ~4.22 Mb; and the largest inversion was identified on the end of chromosome 24, ~12.07 Mb (length estimates are relative to the Clupal_KotzSound assembly and are generally shorter in the *C. harengus* assembly; Supplemental Figure 2). Notably, a large inversion (7.8 Mb) on chromosome 12 in the



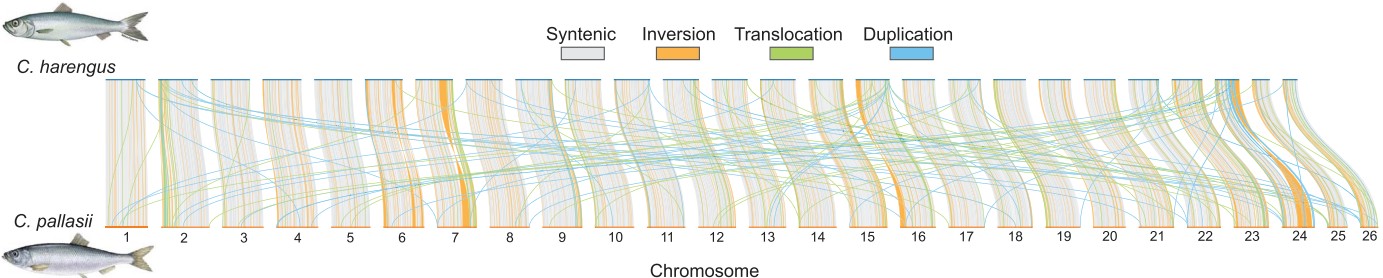

**Figure 1.** Genome wide synteny between *Clupea harengus* and *C. pallasii* KS reveal myriad inversions, duplications, and translocations.

**Table 4.** Summary of SyRi on a minimap alignment of GCF_900700415.2_Ch_v2.0.2_genomic.fna (*Clupea harengus*; as reference) with Clupal_KotzSound_assembly (*Clupea pallasii* KS; as query). Both assemblies were filtered to retain chromosome-length scaffolds (26 chromosomes in each genome) and low divergence between genomes was assumed.

| Annotation type | Variation type | Count | *C. harengus* assembly length | Clupal_KotzSound assembly length |
|---|---|---|---|---|
| **Structural** | Syntenic regions | 1,899 | 577,766,647 | 595,074,621 |
| | Inversions | 407 | 58,519,004 | 66,880,983 |
| | Translocations | 2,073 | 7,117,648 | 7,149,383 |
| | Duplications (reference) | 811 | 3,613,180 | — |
| | Duplications (query) | 2,503 | — | 5,251,103 |
| | Not aligned (reference) | 4,795 | 80,380,302 | — |
| | Not aligned (query) | 6,547 | — | 118,954,211 |
| **Sequence** | SNPs | 2,756,577 | 2,756,577 | 2,756,577 |
| | Insertions | 525,082 | — | 10,626,042 |
| | Deletions | 526,708 | 9,426,962 | — |
| | Copy gains | 781 | — | 750,230 |
| | Copy losses | 646 | 807,757 | — |
| | Highly diverged | 47,449 | 242,958,292 | 267,532,525 |
| | Tandem repeats | 266 | 37,163 | 30,861 |

Atlantic herring genome, characterized as a "supergene" [39, 40], was not identified in the Clupal_KotzSound assembly. The number of duplications detected in the Clupal_KotzSound assembly was more than triple that detected in the *C. harengus* assembly (811 and 2,503, respectively), though chromosomes 5, 12, and 14 lacked duplications in the Clupal_KotzSound assembly. Between the two assemblies, 2,073 translocations were identified. These variants were found on all but one chromosome in the Clupal_KotzSound assembly: no translocations were identified on chromosome 6.

## Phylogeny and divergence

Phylogenetic relationships were estimated from the consensus of maximum likelihood phylogenies built from the nucleotide sequences of 500 protein-coding loci identified in the six publicly available clupeiform assemblies and Clupal_KotzSound (Figure 2). These loci are a subset of the complete single copy genes found across the target assemblies in the BUSCO analysis. Loci were selected based on length (between 1500 and 3000 bp) and length variation among homologs in the target genomes (<5%). A random sample of 500 loci meeting these criteria were selected to estimate phylogenetic relationships and coding sequence divergences. Substitution models and phylogenetic tree topologies were generated in IQ-TREE v3.01 [41–43] and the consensus tree was computed in

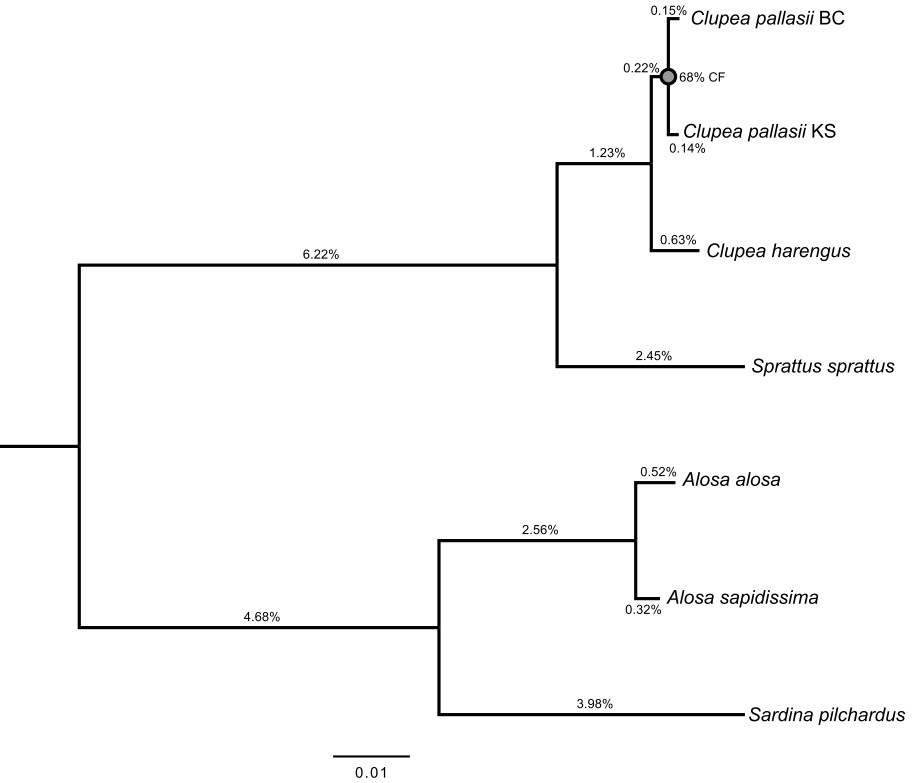

**Figure 2.** Consensus topology of maximum likelihood (ML) phylogenies from 500 protein-coding loci identified in the six publicly available clupeid assemblies and Clupal_KotzSound. Support values reflect the percent of single-locus phylogenies in which the node was recovered. All but one of the relationships in the topology are retrieved in more than 95% of the individual loci phylogenies: only 68% of the individual phylogenies support a sister group relationship between *Clupea pallasii* BC and KS (grey). Branch lengths correspond to median ML lengths calculated from locus-specific optimal substitution models.

DendroPy v5 [44]. All but one relationship in the consensus topology were retrieved in more than 95% of the individual phylogenies: two major clades confidently separated *Clupea* + *Sprattus sprattus* from *Alosa* + *Sardina pilchardus* and monophyly was strongly supported in both *Alosa* and *Clupea*. However, only 68% of the individual phylogenies supported a sister relationship between *Clupea pallasii* BC and *C. pallasii* KS.

Per-site maximum likelihood pairwise genetic distances were also calculated from nucleotide alignments of the same 500 protein-coding loci underlying phylogenetic analysis, determining substitution models and distances independently from locus-specific alignments in IQ-TREE. Overall, pairwise genetic distances ranged from 0.002 (between Pacific herring taxa *C. pallasii* BC and KS) to 0.162 (between *Sprattus sprattus* and *Sardina pilchardus*) (Table 5). Within Genus *Alosa*, *A. alosa* and *A. sapidissima* had a genetic distance twice that of the Pacific herring taxa, but only a quarter larger than those between Pacific herring taxa (BC and KS) and *C. harengus* in the Atlantic (0.003).

**Table 5.** Per site maximum likelihood pairwise distances calculated from nucleotide alignments of 500 protein-coding loci from genome assemblies across seven clupeiform taxa, including *Clupea pallasii* BC and KS. Substitution models and distances were calculated independently from alignments of each locus in IQ-TREE v3.01. Median values above the diagonal and interquartile range below the diagonal.

| | *Clupea pallasii* BC | *Clupea pallasii* KS | *Clupea harengus* | *Sprattus sprattus* | *Alosa alosa* | *Alosa sapidissima* | *Sardina pilchardus* |
|---|---|---|---|---|---|---|---|
| *Clupea pallasii* BC | | 0.0019 | 0.0033 | 0.0343 | 0.1467 | 0.1460 | 0.1521 |
| *Clupea pallasii* KS | 0.0011–0.0032 | | 0.0034 | 0.0342 | 0.1470 | 0.1461 | 0.1528 |
| *Clupea harengus* | 0.0018–0.0058 | 0.0020–0.0060 | | 0.0339 | 0.1471 | 0.1464 | 0.1522 |
| *Sprattus sprattus* | 0.0275–0.0446 | 0.0276–0.0448 | 0.0271–0.0456 | | 0.1547 | 0.1539 | 0.1623 |
| *Alosa alosa* | 0.1227–0.1733 | 0.1224–0.1719 | 0.1231–0.1711 | 0.1282–0.1853 | | 0.0041 | 0.0612 |
| *Alosa sapidissima* | 0.1216–0.1708 | 0.1216–0.1696 | 0.1218–0.1692 | 0.1276–0.1813 | 0.0026–0.0063 | | 0.0597 |
| *Sardina pilchardus* | 0.1275–0.1816 | 0.1272–0.1820 | 0.1277–0.1798 | 0.1343–0.1898 | 0.0491–0.0799 | 0.0479–0.0780 | |

## IMPLICATIONS AND FUTURE WORK

### Cryptic diversity in *Clupea*

Historically, two distinct populations of Pacific herring have been recognized in the Northeast Pacific: a southern population existing south of the Aleutian Islands and a northern population existing north of the Aleutian Islands and in the eastern Bering Sea [45, 46]. Recently, whole genome resequencing revealed genome-wide differentiation between these populations, suggesting these two groups may represent separate species [9]. Results presented here, comparing the scaffold-level assembly from British Columbia with Clupal_KotzSound, provide more clarity on the genomic similarities and differences between northern and southern Pacific herring populations. Moving forward, a taxonomic study of the two populations is needed to elucidate whether they truly represent different species. Preliminary evidence supports the resurrection of *C. mirabilis* (Girard, 1854), described from a type specimen collected from San Francisco, California [47], to represent the Gulf of Alaska taxon.

### Evolutionary history of *Clupea* in the subarctic

*Clupea harengus* and *C. pallasii* have been considered sister species for decades [48, 49] and hybridization following secondary contact in the Northwest Atlantic is well documented [50, 51]. Hypothesized to have separated during the Pliocene, population structure within *C. pallasii* has been attributed to subsequent vicariance as spawning habitat disappeared at the Last Glacial Maximum [9, 46, 52]. However, the phylogenomic analysis of clupeiforms we present, which includes assemblies representing *C. harengus* and both populations of *C. pallasii* in the Northeast Pacific, does not reflect a clear consensus. The lack of a strongly supported terminal node in the tree subtending *C. pallasii* (Figure 2) indicates a need for a more comprehensive analysis of species delimitation within *Clupea*.

### DATA AVAILABILITY

The Clupal_KotzSound (*Clupea pallasii*) reference genome and associated Hi-C and long read sequences can be accessed on NCBI Genbank through accession GCA_056790095.1, BioSample SAMN54268497. Other genomes included in the analyses of this manuscript can be accessed on NCBI either via Genbank or Refseq, with the assembly name, species, and accession listed as follows: fCluPal1_p1.1 (*Clupea pallasii*, GCA_051176545.1) [21], Ch_v2.0.2 (*Clupea harengus*, GCF_900700415.2 on RefSeq, GCA_900700415.2 on Genbank) [20], fSprSpr1.1 (*Sprattus sprattus*, GCA_963457725.1) [24], AALO_Geno_1.1 (*Alosa alosa*, GCF_017589495.1 on RefSeq, GCA_017589495.2 on Genbank) [25], fAloSap1.pri (*Alosa*

*sapidissima*, GCF_018492685.1) [26], fSarPil1.1 (*Sardina pilchardus*, GCF_963854185.1 on RefSeq, GCA_963854185.1 on Genbank) [27], . All additional supporting data [53] are available in the GigaScience repository, GigaDB [54].

## AVAILABILITY SOURCE CODE AND REQUIREMENTS

Project name: Pacific_herring_reference_genome_code

Project homepage:
https://github.com/GlassLabGenomics/pacific_herring_reference_genome_code

Operating system: Linux, MacOS, Windows (with Unix-terminal emulator)

Programming language: Bash

Other requirements: BUSCO v. 6.0.0, AGAT v. 0.7.0, MACSE v. 12.01, DendroPy v. 4.2+, QUAST v. 5.2.0, minimap2 v. 2.24, AGAT v. 0.7.0, RepeatMasker v. 4.2+, SyRI v. 1.7.1, plotsr v. 0.4, IQ-TREE v. 3.01, Python v. 3+, R v. 4+

License: MIT license.

## DECLARATIONS

### Ethics approval

The authors declare that ethical approval was not required for this type of research.

### Competing interests

The authors declare no competing interests.

### Authors' contributions

LET, SAA, JAL, JRG conceptualized the project and acquired funding. LET, SAA, YH generated and curated data. LET, YH, JAL conducted the formal analyses. LET led and all authors contributed to the writing and editing of this article.

### Funding

This research was made possible by the Arctic Fellowship Award, funded by Center ICE at the University of Alaska Fairbanks, the National Science Foundation (Grant No. DBI: 231378) and the Pollock Conservation Cooperative Research Center (PCCRC 22-02 and PCCRC 25-03).

### Acknowledgements

The authors thank the subsistence user who provided the samples used for sequencing.

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
