## [Editor Report]

Editor’s AssessmentThe manuscript is ready for formal acceptance.Editor’s AssessmentThe manuscript is ready for formal acceptance.

---

## [Reviewer Report]

Reviewer name and names of any other individual's who aided in reviewer Mats PetterssonDo you understand and agree to our policy of having open and named reviews, and having your review included with the published papers. (If no, please inform the editor that you cannot review this manuscript.)YesIs the language of sufficient quality?YesPlease add additional comments on language quality to clarify if needed
Are all data available and do they match the descriptions in the paper? NoAdditional CommentsI could not find any records associated with accession JBTGUN000000000. However, the supplied biosample ID (SAMN54268497) is correct, and leads to this project: PRJNA1392659, which contains the expected datasets.Are the data and metadata consistent with relevant minimum information or reporting standards? See GigaDB checklists for examples <a href="http://gigadb.org/site/guide" target="_blank">http://gigadb.org/site/guide</a>YesAdditional CommentsIs the data acquisition clear, complete and methodologically sound?YesAdditional CommentsIs there sufficient detail in the methods and data-processing steps to allow reproduction?YesAdditional CommentsIs there sufficient data validation and statistical analyses of data quality? YesAdditional CommentsIs the validation suitable for this type of data?YesAdditional CommentsIs there sufficient information for others to reuse this dataset or integrate it with other data?YesAdditional CommentsAny Additional Overall Comments to the AuthorExcept for the accession mixup indicated above, I have no critical comments. I am convinced this genome is of high quality, and will prove a useful resource for the scientific community.RecommendationAccept

---

## [Reviewer Report]

Reviewer name and names of any other individual's who aided in reviewer Dr. Shengyong XuDo you understand and agree to our policy of having open and named reviews, and having your review included with the published papers. (If no, please inform the editor that you cannot review this manuscript.)YesIs the language of sufficient quality?YesPlease add additional comments on language quality to clarify if needed
Are all data available and do they match the descriptions in the paper? NoAdditional CommentsThe authors reported that the final assembly consisted of 26 scaffolds, with an N50 of 31 Mb and an L50 of 12; however, the final assembly comprised 322 sequences (GenBank accession JBTGUN000000000.1), including 26 chromosome-level scaffolds and 296 unanchored sequences. The authors should have included information on the unanchored sequences in their statistics to avoid the misunderstanding that the final assembly contains only 26 chromosome-level sequences.Are the data and metadata consistent with relevant minimum information or reporting standards? See GigaDB checklists for examples <a href="http://gigadb.org/site/guide" target="_blank">http://gigadb.org/site/guide</a>YesAdditional CommentsIs the data acquisition clear, complete and methodologically sound?YesAdditional CommentsIs there sufficient detail in the methods and data-processing steps to allow reproduction?NoAdditional CommentsIn the phylogenetic analysis, the authors should describe the process for accessing and filtering 500 protein-coding genes.Is there sufficient data validation and statistical analyses of data quality? YesAdditional CommentsIs the validation suitable for this type of data?YesAdditional CommentsIs there sufficient information for others to reuse this dataset or integrate it with other data?YesAdditional CommentsAny Additional Overall Comments to the Author1. Table 1, GC content data should be rounded to two decimal places, consistent with the description in the main text. 2. Line 3, Page 7, in "Genome assembly validation" section, the scaffold data entries 323, 1496, and 1380 appear to contain mitochondrial genome sequences; the authors are advised to verify this. 3. What do the percentages on the branches of the phylogenetic tree in Figure 2 denote? The authors should provide specific notes. Furthermore, the 68% support level suggests potential taxonomic misidentification; the authors need to clarify the accuracy of the species identification, for example, by conducting DNA barcoding analysis using mitochondrial 12S or COI gene fragments.RecommendationMinor Revision